# Polyphenol-Rich Date Palm Fruit Seed (*Phoenix Dactylifera* L.) Extract Inhibits Labile Iron, Enzyme, and Cancer Cell Activities, and DNA and Protein Damage

**DOI:** 10.3390/nu14173536

**Published:** 2022-08-27

**Authors:** Hosam M. Habib, Esmail M. El-Fakharany, Usama D. Souka, Fatma M. Elsebaee, Mohamed G. El-Ziney, Wissam H. Ibrahim

**Affiliations:** 1Functional Foods and Nutraceuticals Laboratory (FFNL), Dairy Science and Technology Department, Faculty of Agriculture, Alexandria University, Alexandria P.O. Box 21545, Egypt; 2Protein Research Department, Genetic Engineering and Biotechnology Research Institute GEBRI, City for Scientific Research and Technology Applications, New Borg EL Arab, Alexandria P.O. Box 21934, Egypt; 3Department of Nutrition and Health, College of Medicine and Health Sciences, United Arab Emirates University, Al Ain P.O. Box 15551, United Arab Emirates; 4Department of Home Economics, Faculty of Specific Education, Fayoum University, Fayoum P.O. Box 63514, Egypt

**Keywords:** date seed extract, labile iron inhibition, DNA and BSA damage, enzyme inhibition, anticancer

## Abstract

Date palm fruit seed (*Phoenix dactylifera* L.) extract (DSE), an under-utilized resource, is a rich source of polyphenols with high potency for disease prevention and antioxidative activities. For the first time, the present study demonstrated that DSE inhibits labile iron activity and DNA and BSA damage and inhibits acetylcholinesterase and tyrosinase activities. Moreover, DSE reduces the proliferation of hepatic, colorectal, and breast cancer cells dose-dependently through apoptotic mechanisms. Furthermore, DSE significantly suppressed the expression of both BCl-2 and P21 genes and increased the P53 expression level when compared with the untreated cells and the 5-FU treated cells. These findings suggest a strong potential for DSE in protecting against the iron-catalyzed ferroptosis that results in programmed cell death. The results also confirm the efficacy of DSE against cancer cells. Therefore, DSE constitutes a valuable candidate for developing functional foods and for natural compound-based chemotherapy for the pharmaceutical and nutraceutical industries.

## 1. Introduction

According to FAOSTAT, 2020 [1], the global production of date palm fruit (*Phoenix dactylifera* L.) is 9.5 million tons annually. Date palm processing results in the production of vast amounts of date seeds as waste products, constituting 6.11–11.47% of ripe date fruit weight depending on grade quality and the variety [2,3]. Date seeds were used primarily as animal feed and soil fertilizers in the past. However, they have recently been increasingly used in the production of functional foods such as muffins, cookies, gluten-free cookies, cakes, coffee, bread, biscuits, cooking oil, bio-oil, cosmetics and pharmacology, and human dietary supplementation [4,5,6,7,8,9].

Furthermore, date seeds are rich in polyphenols depending on the variety. A concentration of polyphenols of around 4768.87 mg gallic acid equivalents/100 g was reported [10]. We have previously reported that flavan-3-ols are the predominant group of polyphenols in date seeds, which are present as polymeric proanthocyanidins. The polyphenolic content of date seeds is about 51.1 g/kg, which is higher when compared with other polyphenol-rich foods such as grapes, flaxseeds, and tea [11]. In addition, the same study provided clear evidence for the significant bioaccessibility of free polyphenols from date seeds.

The polyphenolic compounds present in DSE have been shown to exhibit health benefits that are important in chronic diseases, regulating metabolism and cell proliferation; however, their short-and long-term health effects have not been thoroughly characterized. Animal, human, and epidemiological studies have indicated that numerous polyphenols have anti-inflammatory and antioxidant properties that could prevent and treat non-communicable disorders, such as cancer, obesity, and neurodegenerative and cardiovascular diseases [12].

DSE polyphenols have been demonstrated to exhibit effective antioxidant activities in vitro and in biological systems. However, to our knowledge, most of the bioactivities of DSE shown in the present study have not been reported earlier. Therefore, the primary objective of this investigation was to be the first to report the functional health effects of DSE on labile iron activity, DNA damage, and protein damage, in addition to the activities of Tyrosinase, Acetylcholinesterase, and Porcine α-amylase, which have been associated with several disease states. Furthermore, the effects of DSE on the proliferation and apoptosis of hepatic, colorectal, and breast cancer cells were investigated.

## 2. Methods

### 2.1. Plant Material and Preparation

Date palm seeds of the Khalas variety were used in this study. Ten kg of dates were randomly collected from fully ripe dates at the end of the harvested season without preference for appearance, color, size, or firmness. First, the date seeds were soaked in normal water, then washed to remove any observed date flesh. Finally, the seeds were dried by air, then ground to powder employing a hammer mill. The seed powder was grounded to a fine powder using a heavy-duty grinder (IKA M 20 Universal Mill; IKA Werke GmbH Co. KG, Staufen, Germany). A Udy cyclone mill was used to allow the powder to pass through a 0.5-mm screen. The fine powder was then separated into two fractions using 0.5 mm and 0.25 mm opening sieves. The powder from the 0.25 mm fraction was extracted by water: methanol (1:1, vol/vol) [10,11]. Date seed powder was mixed with the extraction solvent 1:10 w/vol and checked overnight. The extract was filtered, reduced under nitrogen, and lyophilized.

### 2.2. Study Materials

Khalas variety date seeds were used in this investigation and were donated by the Al Foah Company, Al Ain, UAE. VC.: vitamin C, Rutin, TPTZ: Ferric chloride, Sodium acetate buffer, FeSO_4_, (DPPH^•^): 1,1-diphenyl-2-picrylhydrazyl, sodium nitroprusside, 4-aminobenzenesulfonic acid, anhydrous) acetic acid, N-(1-Naphthyl)ethylenediamine dihydrochloride ACS reagent, sulphuric acid, sodium nitroprusside, sodium phosphate, (NH_4_)2MoO_4_, H_2_O_2_, polysaccharide agarose, [3-(4,5-dimethylthiazol-2-yl)- 2,5 diphenyltetrazolium bromide]: MTT, Dimethyl sulfoxide: (DMSO), trypsin/Ethylenediaminetetraacetic acid, RNase A DNase and protease-free, (EB/AO): PI dye and ethidium bromide/acridine orange, (PFA): paraformaldehyde and pBR322 plasmid DNA were purchased from Millipore Sigma Chemical Co. (Louis, MO, USA). Gibco RPMI-1640 and (DMEM): Dulbecco’s Modified Eagle Medium were purchased from (Thermo Fisher Scientific, Waltham, MA, USA). Normal somatic cells: (HSF), Hepatoma (HepG-2), Colon carcinoma: (Caco-2), and Breast carcinoma: (MDA) cells were purchased from (American Type Culture Collection, Manassas, VA, USA). The biological activities of DSE investigated in this study were compared with the activities of vitamin C and rutin, which are well-established potent antioxidants.

### 2.3. Antioxidant Activity

#### 2.3.1. DPPH^•^-Free Radical-Scavenging Test

The DPPH^•^ activity of Vitamin C (VC), DSE, and rutin was studied at different concentrations from 0.1 mg/mL to 5 mg/mL. They were determined to be effective against 1,1-diphenyl-2, picrylhydrazyl radicals. The results of the study were compared with those of a previously reported method [13]. The percentage inhibition of these radical compounds was determined by taking into account the equation below:% Inhibition DPPH^•^ = (Abs. control − Abs. sample)/(Abs. control) × 100(1)
where the Abs. control test corresponds to the absorbance of the DPPH solution without the test sample. The absorbance was conducted using a Varian CARY 50 Scan UV–VIS Spectrophotometer with a Cary 50 Microplate Reader Accessory (Varian, Inc., Walnut Creek, CA, USA).

#### 2.3.2. ABTS^•^-Free Radical-Scavenging Test

The free radical scavenging capacity of VC, DSE, and rutin was studied at different concentrations from 0.1 mg/mL to 5 mg/mL. They were determined to be effective against ABTS^•^ radicals. The results of the study were compared with those of a previously reported method [14]. The percentage inhibition of these radical compounds was determined by taking into account Equation (1) described above.

#### 2.3.3. Ferric-Reducing/Antioxidant Power (FRAP) Test

The FRAP test was conducted on VC, DSE, and rutin at different concentrations from 0.1 mg/mL to 5 mg/mL, according to the method reported previously [13]. The FRAP reagent proportions consisted of 1:1:10 (*v/v/v*) of TPTZ 10 mM solution in 40 mM HCl, FeCl_3_ 20 mM, and acetate buffer pH 3.6, 0.3 M. Then, 1 mL of the tested sample was mixed with 2 mL of freshly prepared FRAP reagent. The mixtures were incubated for 30 min at 37 °C. The absorbance was conducted using a Varian CARY 50 Scan UV–VIS Spectrophotometer with a Cary 50 Microplate Reader Accessory (Varian, Inc., Walnut Creek, CA, USA), at 593 nm. Deionized water was used as a blank. FeSO_4_ was used for the calibration curve. The values of FRAP were expressed as µmol of Fe (II).

#### 2.3.4. Nitric Oxide Radical (NO) Scavenging Test

NO radical inhibition activity was assessed using the Griess reaction on VC, DSE, and rutin at different concentrations from 0.1 mg/mL to 5 mg/mL, according to the method reported previously [15]. In brief, 3 mL of the reaction mixture (2 mL of 10 mM sodium nitroprusside, 0.5 mL phosphate buffer saline, and 0.5 of tested samples). The mixture was incubated for 150 min at 25 °C. Then 0.5 mL of the mixture was mixed with 1 mL of (sulfanilic acid reagent: 0.33% in 20% glacial acetic acid). After 5 min, 1 mL of naphthyl ethylene diamine dihydrochloride 0.1% was added. After 30 min at room temperature, the absorbance was conducted using a Varian CARY 50 Scan UV–VIS Spectrophotometer with a Cary 50 Microplate Reader Accessory (Varian, Inc., Walnut Creek, CA, USA) at 540 nm. The nitrite percentage was determined by considering Equation 1 described above.

#### 2.3.5. Total Antioxidant Activity

The TAC assay was conducted on VC, DSE, and rutin at (0.1–5 mg/mL) and were investigated according to a previously reported method [15]. In brief, 0.3 mL of a reagent composed of sodium phosphate 28 mM, sulphuric acid 0.6 M, and ammonium molybdate 4 mM were added to 0.1 mL tested samples and incubated at 95 °C for 90 min. After cooling, the mixture was measured at 695 nm using a Varian CARY 50 Scan UV–VIS Spectrophotometer with a Cary 50 Microplate Reader Accessory (Varian, Inc., Walnut Creek, CA, USA). A higher antioxidant capacity is correlated with a higher absorbance value.

#### 2.3.6. Labile Iron Inhibition Test

The effects of Trolox, rutin, gallic acid, and DSE on the release of ferritin-dependent iron superoxide were evaluated using the methods described earlier [16]. Riboflavin 400 mM was mixed with Ferritin 100 mg/mL and methionine 1340 mM, at 10 mg/mL, 50 mg/mL and 100 mg/mL of Trolox, rutin, gallic acid, or DSE. The tested mixture was irradiated with a fluorescent lamp at room temperature for 90 min. Labile iron was measured, as noted previously [16].

### 2.4. Enzyme Inhibitory Activity

#### 2.4.1. Tyrosinase Inhibition Test

The inhibition of the activity of tyrosinase was evaluated as earlier reported [17]. Phosphate buffer 20 mM pH 6.8 was mixed with 4 mL L-tyrosine solution and then added to VC, rutin, DSE, or kojic acid at concentrations from 0.1 mg/mL to 5 mg/mL. After incubation at 37 °C for 10 min, 1 mL 50 units/mL mushroom tyrosinase, dissolved in 0.2 M, pH 6.8 phosphate buffer, was mixed into the mixture. The absorbance was recorded after 10 min at 475 nm using a Varian CARY 50 Scan UV–VIS Spectrophotometer with a Cary 50 Microplate Reader Accessory (Varian, Inc., Walnut Creek, CA, USA). Ethanol 50% was assessed as a blank, while 1 mL of deionized water was assessed as a control. The percentage of tyrosinase inhibition activity was calculated using Equation (1), as described earlier.

#### 2.4.2. Porcine α-Amylase Inhibition Assay

The inhibition activity of porcine α-amylase was evaluated, as reported earlier [17]. Sodium phosphate 0.02 M pH 6.9 was mixed with 50 µL of VC, rutin, DSE, or acarbose at concentrations from 0.1 mg/mL to 5 mg/mL, and sodium chloride 6 mM, contains 13 units/mL α-amylase solution. After incubating the mixture at 25 °C, starch solution mixed with Sodium phosphate 0.02 M pH 6.9 and sodium chloride 6 mM was added and incubated at room temperature for 10 min. After the incubation, 1 mL of dinitrosalicylic acid (color reagent) was added and kept for 10 min at 100 °C to stop the reaction, followed by cooling to 25 °C. Then the absorbance, after adding 1 mL of deionized water, was measured using a Varian CARY 50 Scan UV–VIS Spectrophotometer with a Cary 50 Microplate Reader Accessory (Varian, Inc., Walnut Creek, CA, USA) at 540 nm. The percentage of inhibition activity of porcine α-amylase was calculated using Equation (1), as described earlier.

#### 2.4.3. Acetylcholinesterase Inhibition Test

The inhibition activity of acetylcholinesterase was evaluated as earlier reported [17]. Mixture of 325 µL, 0.05 M, Tris–HCl buffer pH 8, 100 µL of VC, rutin, DSE, or galantamine at concentrations from 0.1 mg/mL to 5 mg/mL and 25 µL, 0.28 U/mL of acetylcholinesterase. At room temperature, the mixture was incubated for 15 min. Then 475 µL 3 mM DTNB solution and 75 µL of acetylcholine iodide 15 mM were added. At room temperature, the mixture was incubated for 30 min. The absorbance of the mixture was measured using a Varian CARY 50 Scan UV–VIS Spectrophotometer with a Cary 50 Microplate Reader Accessory (Varian, Inc., Walnut Creek, CA, USA) at 405 nm. The percentage of inhibition activity of acetylcholinesterase was calculated using Equation (1), as described earlier.

### 2.5. Free Radical-Induced Damage to DNA Test

The DNA damage by the free radical test was conducted using a method outlined earlier [18]. In brief, a prepared mixture of 4 μL of VC, rutin, or DSE at concentrations from 0.1 mg/mL to 5 mg/mL, H_2_O_2_ 30% 6 μL, PBS buffer 6 μL, plasmid pBR322 DNA 0.2 μg dissolved in 2 μL of PBS pH 7.4 50 mM. These mixtures were irradiated using a UV transilluminator TFM-26 (UVP, Upland, CA, USA) for 5 min at 25 °C at the surface of the transilluminator with an intensity of 25 W cm^−2^ at 312. After the reaction, the tested samples were electrophorized by polysaccharide agarose 0.8%. Ethidium bromide was used to stain the electrophoreses gel. The analysis and photographs were performed using Image lab 4.1 software, version 6.1.0. (Bio-Rad, Hercules, CA, USA).

### 2.6. AAPH Induce Protein Oxidation Test

Bovine Serum Albumin (BSA) was oxidized by AAPH using a method outlined earlier [18]. Briefly, 0.5 mg mL BSA was incubated with 20 mM AAPH in the presence or absence of VC, rutin, or DSE at concentrations from 0.1 mg/mL to 5 mg/mL in a water bath with shaking for 30 min at 37 °C. The control sample was without AAPH.

Then the mixture was loaded and separated using SDS-PAGE 10% after reduction conditions for 5 min, at 100 °C. The free stained gels’ images were taken with a ChemiDoc MV gel documentation system (Bio-Rad, Hercules, CA, USA). The band intensity determined the protein damage amount for each band using Image Lab 4.1 Software version 6.1.0 (Bio-Rad, Hercules, CA, USA).

### 2.7. Anticancer Inhibitory Activity

#### Cytotoxicity of DSE on HSF Cells

The cytotoxicity of DSE at different concentrations in the normal cells was evaluated using the method of hydrogen acceptor compared to the (HSF) cells by MTT, as reported previously [19]. Briefly, 5.0 × 10^3^ HSF cells were seeded in a 96-well microplate and cultured by supplemented DMEM with FBS 10% overnight. DSE at different concentrations of 0.0, 100, 200, 300, 400, 500, and 600 μg/mL were incubated in 5% CO_2_ incubator with the cells for 24 and 48 h. After incubation, the cells were washed three times with PBS to remove the dead cells. Then 200 μL of MTT 0.05% stained the cells at 37 °C for approximately 3 to 5 h. After that, the MTT was replaced by 200 μL DMSO. The safe doses EC_10_ and IC_50_ of DSE at 570 nm were measured using a microplate reader (BMG LabTech, Ortenberg, Germany) and software GraphPad Prism version 6.0.

### 2.8. DSE Anti-Cancer Activity by MTT Test

DSE anticancer effect in vitro was estimated by the Caco-2, HepG-2, and MDA cell lines.

The cells were seeded in a 96-well microplate and cultured by supplemented DMEM with FBS 10%, while the HepG-2 cells were maintained in RPMI-1640, with FBS 10%, overnight. DSE at different concentrations of 100, 200, 300, 400, 500, and 600 μg/mL were incubated in a 5% CO_2_ incubator with the cells for 24 and 48 h. The cytotoxicity of DSE was assessed using the MTT test as reported above. The values of SI (selectivity index), which was defined as the ratio of the IC_50_ on HSF cells, ver. the IC_50_ value of each cancer cell line was estimated as reported previously [19]. In addition, the morphology of HepG-2 cells affected by DSE at 100 μg/mL, 200 μg/mL, and 400 μg/mL was investigated using phase-contrast microscopy (Olympus, Germany) and compared to the reference (untreated) cells.

### 2.9. Analysis of Nuclear Staining

PI and EB/AO dye were used to study the apoptotic effect of DSE at concentrations of 100, 200, and 400 μg/mL on HepG-2 cells by fluorescent nuclear staining methods, as reported previously [20].

### 2.10. Analysis of Cell Cycle

The flow cytometry, FACS (Partec, Germany), was used to analyze the HepG-2 cells cycle distribution, which was treated by DSE at a concentration of 100 µg/mL, 200 µg/mL, and 400 µg/mL, as previously reported [20]. The results were calculated using Mod Fit software version 4.0 and Cell Quist software version 3.2. The untreated HepG-2 cells were used as a control.

### 2.11. Quantitative Analysis of Change in the Expression of Oncogenes

The total RNAs were extracted from the untreated and treated HepG-2, Caco-2, and MDA cells using a Gene JET RNA purification kit (Thermo Scientific, Waltham, MA, USA). The cDNA synthesis from mRNA and qPCR was established using a green master mix SYBR with (Forward/Reverse) specific primers. The primers used for BCl2 in this investigation were (5′-TCCGATCAGGAAGGCTAGAGTT-3′/5′-TCGGTCTCCTAA-AAGCAGGC-3′), for p53 (5′-ATGTTTTGCCAACTGGCCAAG-3′/5′-TGAGCAGCGCT-CATGGTG-3′) and p21 (5′-CCACAGCGATATCCAGACATTC-3′/5′-GAAGTCAAAGTTCCACCGTTCTC-3′). The equation (2^−∆∆CT^ (2ˆ (−delta delta of the threshold cycles (CTs).)) was used to calculate the gene expression relative change in the treated and untreated cancer cell lines.

## 3. Statistical Analysis

The statistical analyses were carried out in triplicate (*n* = 3). SPSS for Windows (version 21; SPSS Inc., Chicago, IL, USA) was used to achieve the results using the analysis of variance to evaluate the significance (*p* < 0.05) of the main effects. The different parameter values are stated as mean value ± SD.

## 4. Results and Discussion

### 4.1. Antioxidant Activity

Given the complexity of the antioxidant mechanism, it is always preferable to adopt a multi-method approach to test antioxidant activity. The Khalas variety DSE antioxidant activity was assessed using various assays. This antioxidant activity depends on phenolic compounds’ potential candidates for these activities. The bioactive compound profile of the DSE Khalas variety was detailed in our previous studies [10,11]. The discrepancies observed between our results and other results in the literature are likely due to maturity, fertilizer, season, geographic origin, growing condition, soil type, storage conditions, diseases, and extraction systems [21].

### 4.2. DPPH^•^-Free Radical-Scavenging Assay

DPPH^•^ is a stable free radical that has been used widely to evaluate the effectiveness of natural antioxidants in scavenging free radicals in vitro [13]. Based on the concentrations used in the present investigation, DSE significantly (*p* < 0.05) inhibited the activity of DPPH^•^, as revealed in (Table 1) for IC_50_ and (Figure 1a). The DPPH^•^ scavenging activity of DSE increased dose-dependently with concentrations from 0.1 to 5 mg/mL. In addition, the antioxidant activity of DSE varied between 11.33 ± 0.75% and 21.70 ± 0.06%, with an IC_50_ value of 89.44 ± 3.04 μg/mL. In contrast, the antioxidant activities of vitamin C and rutin (used as comparison standards) ranged from 1.09 ± 0.14% to 67.9 ± 0.85%, with an IC_50_ value of 880 ± 12.89 μg/mL, and 11.49 ± 0.49% to 40.72 ± 0.90% with an IC_50_ value of 415.90 ± 5.38 μg/mL, respectively. Nevertheless, our results were in the same range as those reported previously [22,23].

### 4.3. ABTS^•^-Free Radical-Scavenging Assay

The ABTS^•^ assay is widely used to measure the ability of antioxidants to eliminate ABTS^•^ radicals in vitro [24]. As illustrated in (Figure 1b) and (Table 1), DSE, vitamin C, and rutin were highly capable of scavenging ABTS^•^. This ranged from (86.53 ± 1.22% to 90.45 ± 0.06% with an IC_50_ value of 3.19 ± 0.51 μg/mL) for DSE, (88.29 ± 0.22% to 90.13 ± 0.04% with an IC_50_ value of 0.68 ± 0.12 μg/mL) for vitamin C, and (41.67 ± 0.94% to 92.69 ± 0.32 with an IC_50_ values 6.31 ± 1.00 μg/mL) for rutin. The presence of hydroxylated compounds can explain the ability of DSE to scavenge ABTS^•^ radicals [25]. Nevertheless, our results showed higher activity than that reported previously [23].

### 4.4. Ferric-Reducing/Antioxidant Power Assay

This assay is based on reducing a ferric tripyridyl triazine complex to its colored ferrous form in the presence of antioxidants. As presented in (Figure 1c) and (Table 1), DSE, vitamin C, and rutin reduced the ferric 2,4,6-tripyridyl-s-triazine complex [Fe (III)-(TPTZ)2]^3+^ to [Fe (II)-(TPTZ)2]^2+^, an intensely blue-colored ferrous complex in an acidic medium [13]. DSE, VC, and rutin ranged from (369.23 ± 14.90 to 2087.46 ± 43.03 µmol ferrous equivalents with IC_50_ values 14,191 ± 22.97 μg/mL), (1151.67 ± 8.02 to 10,233.67 ± 243.72 µmol ferrous equivalents with IC_50_ values of 5106 ± 40.54 μg/mL), and (1705.00 ± 15.00 to 10,644.00 ± 86.02 µmol ferrous equivalents with IC_50_ values of 533 ± 22.85 μg/mL), respectively. Bioactive compounds can also act as metal chelators, preventing the oxidation produced by highly reactive hydroxyl radicals. This assay shows DSE displayed the lowest reducing power compared with VC and rutin. Nevertheless, our results were in the same range as previously reported [23].

### 4.5. Nitric Oxide Radical Scavenging Assay

The overexpression of inducible nitric oxide synthase (iNOS) is induced by the activation of immune cells, such as neutrophil granulocytes and macrophages, by lipopolysaccharide (LPS) during bacterial infection. It is also induced by cytokine production, such as IL-1, TNF-α, or IFN-γ. The induction of iNOS enhances the overproduction of nitric oxide (NO), which plays a crucial role in the development of inflammation. In vitro assays prove that non-steroidal anti-inflammatory drugs can block iNOS and scavenge nitric oxide. Moreover, several studies showed that NO enhances the expression of COX-2, resulting in increased formation of prostaglandins. Therefore, the ability to scavenge NO may be crucial for the therapeutic management of inflammatory diseases [26].

DSE and rutin ranged from (14.32 ± 0.23 to 61.81 ± 0.24 with IC_50_ values of 427.70 ± 63.33 μg/mL) and (59.42 ± 0.10 to 79.01 ± 0.07 with IC_50_ values of 270.10 ± 18.62 μg/mL), respectively, for NO inhibition, as shown in (Table 1) and (Figure 1d). The VC scavenging capacity for NO started with a value of 59.10%, reduced to 40.92% and 29.57% at 0.5 mg/mL and 1 mg/mL, respectively, then increased to 55.12% and 56.55% at 3 mg/mL and 5 mg/mL, respectively, with IC_50_ values of 2710 ± 65.98 μg/mL, this may be explained by the antioxidant–prooxidant-dependent mechanisms of VC at different concentrations. The ability of DSE to quench NO suggests a strong potential for protecting against the harmful consequences of extreme NO production. Furthermore, the harmful reactions induced by NO overproduction may be inhibited by the scavenging activity of DSE [26].

### 4.6. Total Antioxidant Activity

The total antioxidant activity (TAC) method is based on the ability of antioxidants to reduce Mo (VI) to Mo(V), which in turn reacts with phosphate producing a green phosphate/Mo(V) complex. DSE, VC, and rutin showed increasing OD with increasing concentrations with green color formation, revealing the scavenging activity of DSE, VC, and rutin towards molybdate. The increasing TAC with increasing pigment concentration indicates an excellent antioxidant property of the pigment. (Table 1) and (Figure 1e) show that DSE, VC, and rutin vary from (0.12 ± 0.00 to 0.88 ± 0.05 with an IC_50_ of 2983.26 ± 22.93 μg/mL), (0.047 ± 0.000 to 0.041 ± 0.000 with an IC_50_ value of 3005.29 ± 427.80 μg/mL), and (0.13 ± 0.00 to 1.57 ± 0.01 with an IC_50_ value of 13,002 ± 213.60 μg/mL), respectively.

### 4.7. Labile Iron Inhibition Assay

Although iron is vital for many critical metabolic processes in the human body, iron accumulation driven by excessive generation of the superoxide-free radical can have detrimental effects on the cell [27]. Labile unbound iron catalyzes the generation of the extremely reactive and toxic hydroxyl radical by the Fenton and Haber–Weiss reactions [16]. Labile iron also causes the production and accumulation of lipid hydroperoxides, which may result in programmed cell death ferroptosis [27]. We have demonstrated in an earlier study that the antioxidants rutin, Trolox, and gallic acid can inhibit the release of ferritin iron by superoxide dose-dependently [27]. Hence, the objective of the current study is to investigate if the polyphenol-rich DSE exerts the same effects. The results presented in (Figure 1f) and (Table 1) show that DSE similarly reduced iron release from ferritin. The percent inhibition by DSE was 68.69 ± 0.30, 79.31 ± 4.13, and 87.47 ± 0.63%, at 10, 50, and 100 μg, respectively, with an IC_50_ value of 4.92 ± 0.65 μg/mL Rutin had 85.32 ± 0.56, 90.30 ± 0.01, and 96.65 ± 0.01% inhibition, with an IC_50_ value of 0.08 ± 0.14 μg/mL, whereas that for gallic acid was 50.68 ± 0.18, 59.22 ± 0.81, and 68.55 ± 0.38% with an IC_50_ value of 4.86 ± 0.23 and that for Trolox was 39.33 ± 0.54, 84.79 ± 3.00, and 87.87 ± 1.87% with IC_50_ value of 12.57 ± 0.64, at 10, 50, and 100 μg, respectively. The differences observed could be attributed to the effectiveness of the antioxidants in scavenging superoxide and their iron-chelating capacities [16].

### 4.8. Enzyme Inhibitory Activity

#### Tyrosinase Inhibition Assay

Tyrosinase is a copper-containing enzyme that produces melanin in hair and skin and contributes to the formation of dopaquinone. The unnecessary generation of melanin contributes to diseases such as Parkinson’s disease and skin cancer [17]. As shown in (Table 1) and (Figure 2a), the percent inhibition of tyrosinase by GSE, VC, and rutin ranged from (37.6 ± 0.87 to 84.33 ± 0.11 with an IC_50_ value of 472.90 ± 1.00 μg/mL), (68.43 ± 0.22 to 98.62 ± 0.03% with an IC_50_ value of 42.74 ± 0.23 μg/mL), (31.80 ± 0.36 to 54.48 ± 1.01% with an IC_50_ value of 72.08 ± 0.54 μg/mL) at 0.1 to 5 mg/mL, respectively, and inhibition by the positive control kojic acid ranged from 79.77 ± 0.32 to 100 ± 0.00% with an IC_50_ value of 24.25 ± 2.09 at 0.1–5 mg/mL. Our results showed the same trend of tyrosinase inhibition activity as the data reported previously [8,23]. The hydroxyl groups present in the polyphenolic compounds of DSE might cause the inhibition reaction through binding via hydrogen bonding to sites of the enzyme, resulting in lower enzyme activity. Moreover, due to their polyhydroxy phenolic structures, flavonoids may chelate copper ions found in the tyrosinase active site, inhibiting its activity. Polyphenolic compounds are effective inhibitors of the activity of this enzyme [17,28,29]. Furthermore, Yu et al. (2019) and Yu & Fan (2021) [28,29] reported that ferulic acid and cinnamic acid, which are found among DSE polyphenols, inhibit tyrosinase activity effectively and bind to various sites of tyrosinase and inhibit tyrosinase activity synergistically.

### 4.9. Porcine α-Amylase Inhibition Assay

Amylase is involved in digesting starch into smaller oligosaccharides and simple carbohydrates. Hence, compounds that inhibit this enzyme may reduce postprandial hyperglycemia, which is potentially therapeutic for those with type 2 diabetes and obese individuals [14,17]. As shown in (Table 1) and (Figure 2b), the percent inhibition of α-amylase by DSE, VC, acarbose (positive control), and rutin ranged from (24.57 ± 0.95 to 84.98 ± 0.55% with an IC_50_ value of 434.80 ± 0.85 μg/mL), (46.13 ± 0.35 to 92.93 ± 0.18% with an IC_50_ value of 87.24 ± 0.71 μg/mL), (44.12 ± 0.20 to 99.92 ± 0.12% with an IC_50_ value of 223.6 ± 18.59 μg/mL) and (46.05 ± 0.52 to 91.51 ± 0.51% with an IC_50_ value of 79.60 ± 0.55 μg/mL), respectively. DSE Polyphenols and flavonoids may be responsible for inhibiting α-amylase, offering potential treatments for hyperglycemia. DSE notably inhibited porcine α-amylase, which was likely due to hydrogen bonding formation between the hydroxyl groups present in polyphenols and the active site of the enzyme [30]. Polyphenols bind to porcine α-amylase and disrupt the secondary structure of the enzyme via the formation of hydrogen bonding. This hampers the substrate channeling through the active site, resulting in the inactivation of the enzyme [30].

Our data were higher than those previously reported [22] and lower than those previously reported [31].

### 4.10. Acetylcholinesterase Inhibition Assay

Acetylcholinesterase inhibition activity significantly attenuates neurodegenerative diseases such as Alzheimer’s disease (AD) pathology based on the mediating analgesic effects [17]. AD treatment is based mainly on available medicines, such as (donepezil, rivastigmine, and galantamine). Nevertheless, these medicines have limitations because of their unfavorable side effects, short half-lives, problems associated with bioavailability, toxicity, and gastrointestinal disturbances, while natural inhibitors of acetylcholinesterase may be effective in Alzheimer’s disease treatment but with no side effects [17]. Table 1 and (Figure 2c) show that the percent inhibition of the enzyme by DSE, VC, galantamine (positive control), and rutin ranged from (39.55 ± 1.07 to 72.15 ± 1.27% with an IC_50_ value of 92.46 ± 1.18), (3.39 ± 0.28 to 6.89 ± 0.34% with an IC_50_ value of 1113 ± 0.34 μg/mL), (52.54 ± 0.21 to 100 ± 0.00% with an IC_50_ value of 102.9 ± 9.01 μg/mL) and (5.53 ± 0.63 to 15.03.51 ± 0.86% with an IC_50_ value of 421.70 ± 0.60 μg/mL), respectively. The inhibition mechanism(s) might be due to the interaction of polyphenols with the enzyme’s active site, as discussed above for α-amylase and tyrosinase. Some studies provide evidence of acetylcholinesterase inhibition by DSE [8,32].

### 4.11. DNA Damage by Free Radicals

The protection of DSE against the oxidative damage of the pBR322 plasmid DNA triggered by the highly mutagenic UV light and H_2_O_2_ was determined. The results of this experiment are displayed in (Table 1) and (Figure 3a–c). DNA strand scission was caused by the hydroxyl radical (OH^•^) produced by the UV light photolysis of H_2_O_2_. The degradation of the plasmid DNA by the hydroxyl radical resulted in three bands; a slow-moving open circular (OC) band, a rapid-moving native supercoiled (SC) band, and a linear form band. DSE, vitamin C, and rutin reduced DNA damage. The untreated plasmid showed 92.70 ± 2.78% SC and 7.30 ± 2.78% OC, while the plasmid treated with UV and H_2_O_2_ exhibited 100% OC. The results showed that pretreatment with DSE, rutin, or vitamin C protected against DNA damage, as shown by the elevation of the supercoiled form and a reduction in the open circular structure of DNA. The results of DSE for the supercoiled DNA ranged from 2.73 ± 0.26 to 4.67 ± 0.41%, and for the linear form ranged from 5.92 ± 0.04 to 7.86 ± 0.08%, with an IC_50_ value of 48.46 ± 2.00 μg/mL, while those for vitamin C ran from 11.50 ± 0.63 to 19.10 ± 0.91% with an IC_50_ value of 42.03 ± 1.06 μg/mL, and those for rutin went from 13.50 ± 0.52 to 66.50 ± 0.57% with an IC_50_ value of 2.45 ± 0.13 μg/mL, at 0.1 to 5 mg/mL, respectively. Rutin showed the most protective effect against the damage of pBR322 plasmid DNA, followed by vitamin C and DSE. For the first time, the results showed that DSE effectively protected DNA from ionizing radiation in vitro, independent of cellular defense and DNA repair systems. The protective effect of DSE against DNA damage in the pBR322 system may be attributed to its ability to quench O_2_^•−^ and (OH^•^).

### 4.12. Protein Oxidation Induced by AAPH

Protein oxidation by free radicals leads to the covalent alteration of the protein molecules, which results in functional changes in the protein. The results of the BSA electrophoresis and the corresponding densitometric analyses of related bands after incubation with AAPH for 30 min with or without different DSE, rutin, or vitamin C concentrations are shown in (Table 1) and (Figure 4a–c). While the band density of control BSA (lane 1) was 100%, the treated BSA (lane 2) band density reduced to 18.98 ± 0.32% following the incubation with AAPH for 30 min. Treatment with DSE, vitamin C, or rutin (lanes 3–7) with concentrations from 0.1 to 5 mg/mL resulted in protective effects against BSA damage. BSA fragmentation decreased by 94.61 ± 0.26 to 95.14 ± 0.06% for DSE with an IC_50_ value 1.17 ± 0.20 μg/mL, while those for vitamin C ran from 70.13 ± 0.34 to 81.34 ± 0.12% with an IC_50_ value 25.94 ± 0.28 μg/mL, and those for rutin went from 94.12 ± 0.20 to 99.10 ± 0.24% with an IC_50_ value 4.42 ± 0.24 μg/mL at 0.1 to 5 mg/mL, respectively. However, vitamin C exhibited antioxidant-prooxidant effects at doses of 3 mg/mL and 5 mg/mL, respectively. This is the first report showing the protective effects of DSE against free radical-induced protein damage.

### 4.13. Anticancer Inhibitory Activity

#### In Vitro Cytotoxicity Effects of the DSE

The viability of DSE was assayed by the MTT assay to determine its cytotoxicity against cancer cells and its safety in normal cells. Herein, we evaluated the cytotoxicity of DSE on the HSF (normal), HepG-2 (liver cancer), Caco-2 (colon cancer), and MDA (breast cancer) cell lines to determine its selectivity. The highest safety is indicated by the highest EC_100_ and IC_50_ values. The data in (Table 1) show that the EC_100_ and IC_50_ values for DSE were 1.85–2.1 and 2.2–2.95 times higher in normal cells compared with cancer cells after 24 h and 48 h of treatment, respectively. (Table 1) also reveals that the DSE had potent antitumor activity against HepG-2, Caco-2, and MDA cells at IC_50_ values of 282 ± 13.56, 321.1 ± 10.28, and 312.9 ± 7.89 μg/mL, respectively, with SI values of 2.1 ± 0.17, 1.85 ± 0.06, and 1.89 ± 0.03, respectively, after treatment for 24 h. However, the values of IC_50_ were found to be decreased with increasing the time of treatment for 48 h, which were determined to be 191.3 ± 8.16, 256.6 ± 12.23, and 219.1 ± 10.14 μg/mL with SI values of 2.95 ± 0.14, 2.2 ± 0.08, and 2.58 ± 0.11 for the HepG-2, Caco-2, and MDA cells, respectively. Furthermore, (Figure 5A(I,II)) indicates that DSE had a dose-dependent cytotoxic effect, with increased safety of normal HSF cells and increased selectivity against cancer cells.

In addition, the cell cycle arrest distribution (I) of HepG-2 of the DSE-treated cells was examined to gain insight into its anticancer mechanism. (Figure 5B(II)) indicates that DSE enhances cell cycle arrest and the distribution of cell population growth in G0/G1 and G2/M (the main checkpoint phases). The results showed a clear increase in the percentages of sub-G1phase by increasing the dose of treatment (100, 200, and 400 μg/mL). However, the synthesis (S) phase diminished in a dose-dependent manner, as shown in (Figure 5B). Our findings demonstrate the ability of DSE to induce cell cycle arrest in treated hepatoma cells compared with untreated cells.

Furthermore, (Figure 5C) indicates that DSE can downregulate oncogenes (BCl-2 and P21) and upregulate the pro-apoptotic gene (P53) in the treated HepG-2 (I), Caco-2 (II), and MDA (III) cell lines. It was found that the DSE (IC_50_ dose) has a significantly higher effect on the suppression of the expression of both BCl-2 and P21 genes and increases P53 expression level by more than 2–8 folds compared to the untreated cells and 2–3 folds compared to the 5-FU treated cells.

Figure 6 shows the results of the morphological changes and nuclear staining of the HepG-2 cells after treatment with DSE (100, 200, and 400 μg/mL) for 48 h. All photomicrographs revealed that the morphology of the treated cells was significantly changed due to the treatment (Figure 6A). The morphological changes proved that DSE induced noticeable cell destruction and altered the cell morphology in a dose-dependent manner. These changes included nuclear condensation, cell shrinkage, and blabbing (Figure 6A). Another advance to prove the apoptotic effect of DSE was carried out by capturing the nuclear alternation in the HepG-2 cells after treatment. (Figure 6B) shows that treated cells’ nuclei became progressively brilliant fluorescent with increasing doses of DSE, in addition to condensation and chromatin fragmentation, compared to the untreated HepG-2 cells, which showed negligible PI-positive staining. In addition, AO/EB fluorescent nuclear staining shows that the apoptotic effect occurred late in the DSE-treated HepG-2 cells, which lost the integrity of their membranes and emitted orange fluorescence as opposed to green fluorescence in negative reference cells (Figure 6C). Moreover, necrotic cells increased dose-dependently, and the nuclei of treated cells emitted uneven red fluorescence instead of green fluorescence at their periphery. The volume of necrotic cells increased, and the cells appeared to be in the process of decomposition (Figure 6C).

Natural products are important for maintaining human health and are good bioprospecting sources for exploring new antitumor drugs. Therefore, the discovery of new antitumor candidates from low-cost and affordable sources is in continuous demand. Al-Sheddi (2019) [33] demonstrated the antitumor properties of seed extract from *Phoenix dactylifera* against liver, lung, and breast cancer cells. In addition, Rezaei (2015) [34] reported the anticancer potential of DSE against colorectal cancer cells. Another study revealed the anticancer potential of DSE against breast and liver cancer cells [35]. Date seeds are rich in polyphenols, phenolic acids such as protocatechuic acid, caffeoyl shikimic acids, and flavan-3-ols compounds, including catechin, epicatechin, and polymeric proanthocyanidins. Natural polyphenols have previously been shown to inhibit several important biochemical routes contributing to cancer development [36]. Additionally, polyphenols may protect living cells from ROS damage and boost immunity. In a previous study, Habib et al. (2014) [37] documented the anticancer activity of date seed against pancreatic cells. Hilary et al., (2021) [35], found that date seed enhanced the anticancer activity, but a high concentration of more than 1000 μg/mL was used with IC_50_ values of 678.4 μg/mL and 662.2 μg/mL on MCF-7 and HepG-2 cells, respectively, after an exposure time of 48 h. However, Al-Zubaidy, (2016) [38] revealed the anticancer activity at a low concentration of polyphenols extracted from date seed with chloroform against MCF-7 cells with IC_50_ determined to be 3.18 μg/mL after 48 h of treatment. Recently, Khan et al. (2022) [39] demonstrated that DSE exhibits anticancer activity against the MDA-MB-231, MCF-7, and HepG-2 cells with IC_50_ values of 85.86, 99.9, and 122.3 μg/mL after treatment for 48 h, respectively. Furthermore, the DSE enhanced the expression level of p53 and suppressed the expression levels of both p21 and BCl-2. The gene of BCl-2 is an anti-apoptotic gene that inhibits the initiation steps of the apoptosis pathway and programmed cell death [40]. However, p53 is a crucial regulatory protein in the enhancement of apoptosis that is mediated after DNA damage by anticancer drugs [41], which consequently leads to the arrest of cell cycle growth in the apoptosis phase or sub-G1 phase. Therefore, our results suggested that the date seed extract had a higher induction of apoptosis in all treated cancer cells. Compatible with our results, a recent study has revealed that the date seed extract exerted anticancer activity against human breast and liver cancer cells by apoptosis signaling pathways, including BCl-2, Bax, p53, and cleaved caspase-3 and PARP-1 in breast cancer cells treated with date seed extract [39]. They revealed that date seed could deactivate BCl-2 protein and activate Bax and Bak form homo-oligomers, which causes pore formation and disturbs the permeability of the outer membrane of mitochondria. This releases the content of the mitochondrial inner membrane, such as Smac and Cyt c, into the cytosol. However, they revealed that the DSE did not reduce p53 protein, which indicates that DSE could induce the apoptotic effect through the p53-independent route. Furthermore, Khan et al. (2022) [39] have reported an anticancer effect of date seed-mediated apoptosis related to arresting the G1/S cell cycle, leading to suppression of BCl-2 level on the treated cancer cell lines.

## 5. Conclusions

It has become evident that DSE is a rich source of highly therapeutic bioavailable polyphenolic compounds [10,11]. Therefore, the primary emphasis of current research conducted on DSE is to explore its health benefits and the underlying mechanisms of action. The results from the present investigation demonstrate, for the first time, that DSE inhibits free radicals and labile iron activities, and DNA and protein damage, suggesting a strong potential for DSE in protecting against oxidative damages and the programmed cell death triggered by iron-catalyzed ferroptosis. Furthermore, DSE also inhibited acetylcholinesterase, α-amylase, and tyrosinase, whose activities have been associated with several widespread diseases. The present study provides additional evidence for the role of DSE in bringing about the death of hepatic, colorectal, and breast cancer cells by apoptosis. These findings should support the use of DSE in functional foods and nutraceutical products and stimulate in vivo and clinical trials involving DSE.

## Figures and Tables

**Figure 1 nutrients-14-03536-f001:**
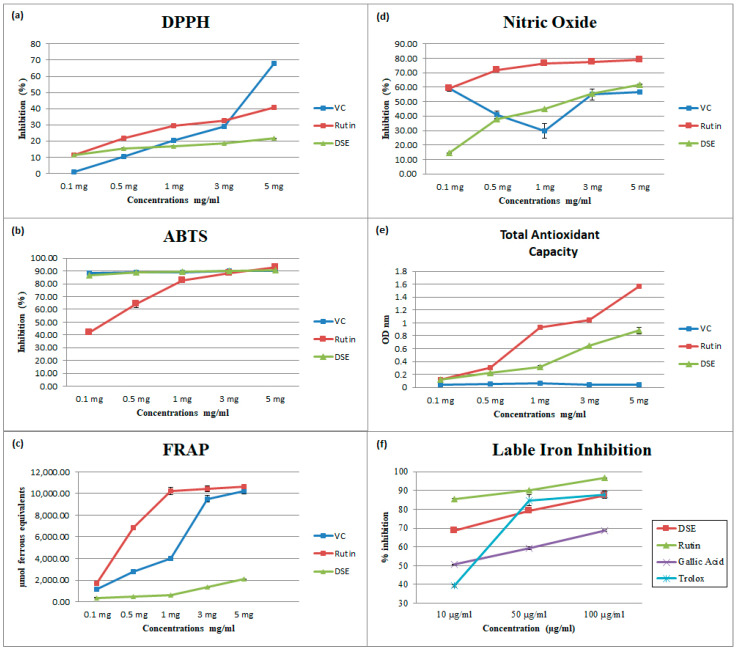
(**a**) 2,2-diphenyl-1-picrylhydrazyl, (DPPH^•^), (**b**) 2,2’-azino-bis(3-ethylbenzothiazoline-6-sulfonic acid, (ABTS^•^), (**c**) Ferric-Reducing/Antioxidant Power (FRAP), (**d**) Nitric Oxide, and (**e**) Total Antioxidant Capacity assays for date seed extract, (DSE), vitamin C, and rutin at different concentrations. (**f**) Effects of DSE, gallic acid, rutin, and Trolox on inhibition of iron release from ferritin by superoxide. Data are expressed as the mean ± SD; *n* = 3.

**Figure 2 nutrients-14-03536-f002:**
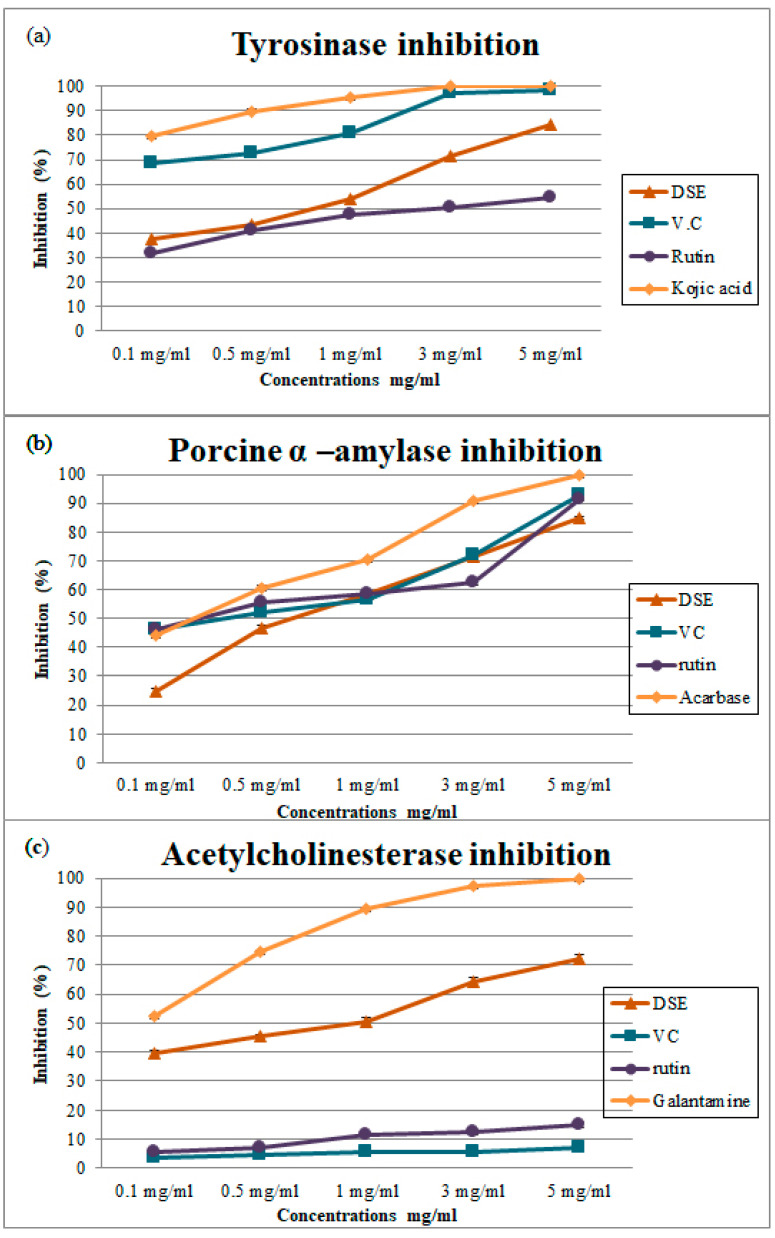
(**a**) Tyrosinase inhibitory activity, (**b**) Porcine α-amylase inhibitory activity, and (**c**) Acetylcholinesterase inhibitory activity for DSE, rutin, and vitamin C at different concentrations. Data are expressed as the mean ± SD; *n* = 3.

**Figure 3 nutrients-14-03536-f003:**
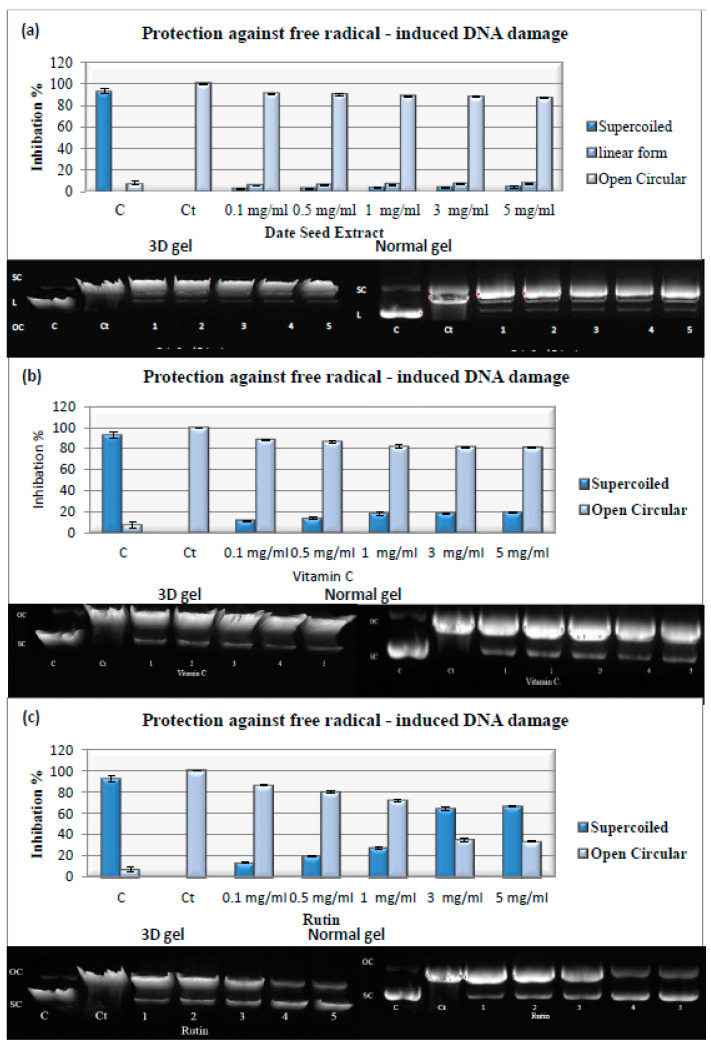
Normal gel, 3D gel, and densitometric analysis for (**a**) DSE, (**b**) V.C., and (**c**) rutin at different concentrations. C: Plasmid, Ct.: plasmid + H_2_O_2_ + UV, Lanes 1–5: plasmid + DSE, V.C., or rutin at 0.1–5 mg/mL + H_2_O_2_ + UV. C and Ct are cropped from different parts of the same gel. Data are expressed as the mean ± SD; *n* = 3.

**Figure 4 nutrients-14-03536-f004:**
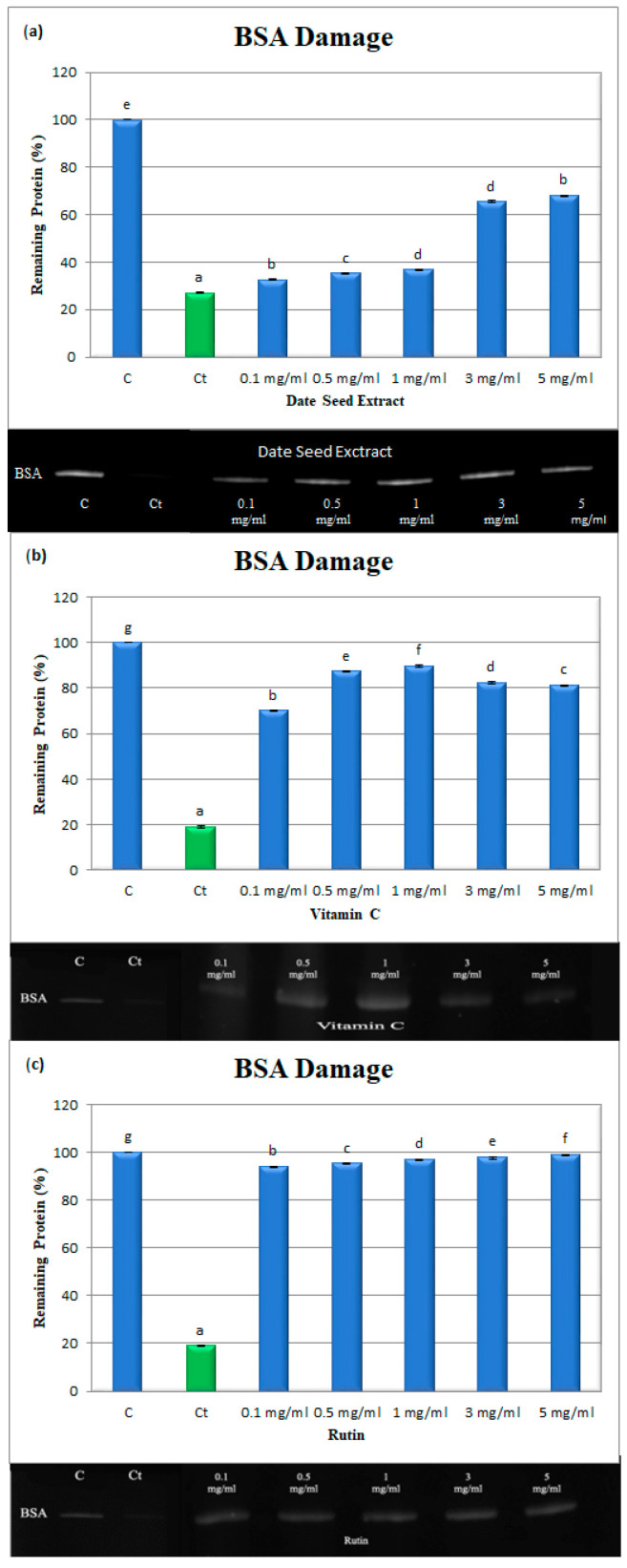
Densitometric analysis and SDS-PAGE of the effects of (**a**) DSE, (**b**) V.C., and (**c**) rutin, on the oxidative damage of BSA. C: BSA, Ct: BSA + AAPH, BSA + AAPH + DSE, V.C., or rutin at concentrations of 0.1–5 mg/mL. C and Ct are cropped from a different gel Data are expressed as the mean ± SD; *n* = 3. Different letters in a column denote significant differences, *p* < 0.05.

**Figure 5 nutrients-14-03536-f005:**
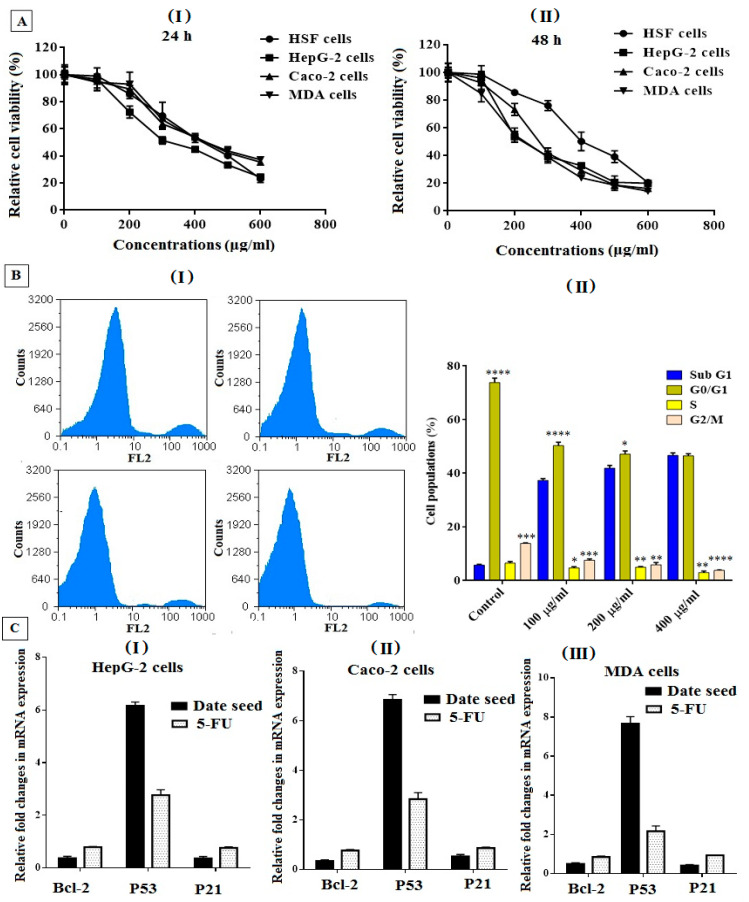
(**A**) DSE effect on the viability of cancer (MDA, HepG-2, and Caco-2) and normal (HSF) cell lines using the MTT method. All cell lines were incubated with DSE at different concentrations (0–600 μg/mL) for 24 h (**I**) and 48 h (**II**). All values are expressed as mean ± SEM; *n* = 3. (**B**) Cell cycle distribution of DSE-treated HepG-2 cells at different doses for 48 h, (**I**) cell cycle distribution diagrams (original flow charts), and (**II**) quantitative distribution of the cells treated with DSE in different phases of the cell cycle compared with untreated (control) cells. All values are demonstrated as mean ± SEM, and differences were considered statistically significant at * *p* < 0.05, ** *p* < 0.005, *** *p* < 0.0005, and **** *p* < 0.0001. (**C**) Relative fold change in the gene expression of BCl2, P53, and p21 in the DSE treated cells using qPCR. Angiogenesis-related genes are evaluated in HepG-2 (**I**), Caco-2 (**II**), and MDA (**III**) cells before and after treatment with the DSE compared with 5-FU for 48 h. Values are presented as mean ± SEM.

**Figure 6 nutrients-14-03536-f006:**
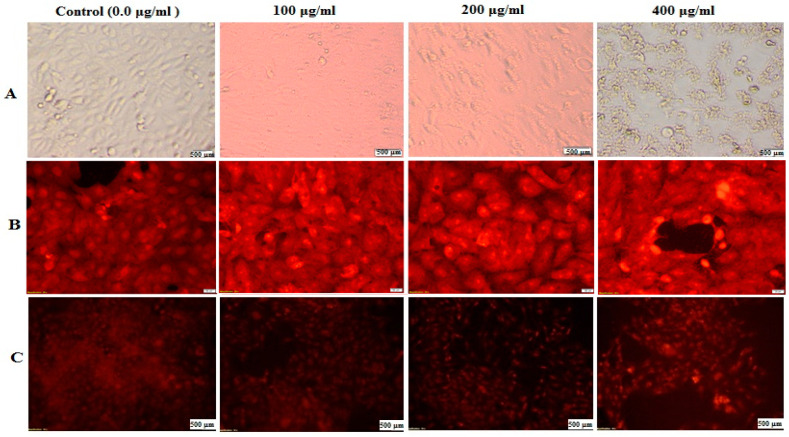
DSE effect at different concentrations of 0.0 μg/mL (control), 100 μg/mL, 200 μg/mL, and 400 μg/mL on HepG-2 cells using an inverted phase-contrast microscope: (**A**) Morphological modifications of the cells treated with DSE, (**B**) Fluorescence images of PI-stained cells, and (**C**) Fluorescence images of HepG-2 cells stained with ethidium bromide–acridine orange.

**Table 1 nutrients-14-03536-t001:** Effects of DSE, vitamin C, and rutin on (A) Free radical inhibition activity, (B) Enzyme inhibition activity, (C) DNA, BSA damage inhibition activity, and (D) DSE: EC_100_, IC_50_, and SI against HSF, HepG-2, Caco-2, and MDA cell lines after treatment for 24 h and 48 h.

(A) Free Radical Inhibition Activity—IC_50_ (µg/mL)											
DPPH				ABTS				FRAP			
DSE	VC	Rutin		DSE	VC	Rutin		DSE	VC		Rutin
89.44 ± 3.04	880 ± 12.89	415.90 ± 5.38		3.19 ± 0.51	0.68 ± 0.12	6.31 ± 1.00		14,191 ± 22.97	5106 ± 40.54		533 ± 22.85
Total antioxidant Capacity					Nitric oxide			Labile Iron			
DSE	VC	Rutin		DSE	VC	Rutin		DSE	Rutin	Gallic acid	Trolox
2983.26 ± 22.93	3005.29 ± 427.80	13,002 ± 213.60		427.70 ± 63.33	2710 ± 65.98	270.10 ± 18.62		4.92 ± 0.65	0.08 ± 0.14	4.86 ± 0.23	± 0.64
(B) Enzyme Inhibition Activity—IC_50_ (µg/mL)											
Tyrosinase				Porcine α-amylase				Acetylcholinesterase			
DSE	VC	Rutin	Kojic acid	DSE	VC	Rutin	Acarbose	DSE	VC	Rutin	Galantamine
472.90 ± 1.00	42.74 ± 0.23	72.08 ± 0.54	24.25 ± 2.09	434.80 ± 0.85	87.24 ± 0.71	79.60 ± 0.55	223.6 ± 18.59	92.46 ± 1.18	1113 ± 0.34	421.70 ± 0.60	102.9 ± 9.01
(C) DNA, BSA Damage Inhibition Activity—IC_50_ (µg/mL)											
DNA					BSA						
DSE		VC		Rutin	DSE			VC	Rutin		
48.46 ± 2.00		42.03 ± 1.06		2.45 ± 0.13	1.17 ± 0.20			25.94 ± 0.28	4.42 ± 0.24		
(D) DSE: EC_100_, IC_50_, and SI (μg/mL)											
		HSF		HepG-2			Caco-2	MDA			
		24 h.	48 h.	24 h.	48 h.	24 h.		48 h.	24 h.		48 h.
EC_100_		59.65 ± 2.12	43.05 ± 1.72	29.07 ± 2.93	21.35 ± 1.76	32.12 ± 1.03		25.47 ± 2.64	55.66 ± 2.5		21.92 ± 2.19
IC_50_		592.6 ± 21.07	565 ± 17.11	282 ± 13.56	191.3 ± 8.16	321.1 ± 10.28		256.6 ± 12.23	312.9 ± 7.89		219.1 ± 10.14
SI		-	-	2.1 ± 0.17	2.95 ± 0.14	1.85 ± 0.06		2.2 ± 0.08	1.89 ± 0.03		2.58 ± 0.11

HSF: normal somatic cells; HepG-2: liver cancer; Caco-2: colon cancer; and MDA: breast cancer cell lines. All values are expressed as mean ± SD.

## Data Availability

Not applicable.

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
