# Peer review of "Polyphenol-Rich Date Palm Fruit Seed (Phoenix Dactylifera L.) Extract Inhibits Labile Iron, Enzyme, and Cancer Cell Activities, and DNA and Protein Damage"

_nutrients, 2022, doi:10.3390/nu14173536_

Round 1

Reviewer 1 Report

In general, the theme approached by the authors in this study is very interesting and topical. the research of nutraceuticals is a subject that deserves more investigation because of its interest for the sustainable human development. However, this work in this form does not facilitate the comprehension of the thematic.

The introduction can be improved;

The authors should review the way they cite figures in the body of the text;

The bibliographic references should be revised according to the standards of the journal.

Authors can use a recent article published in "Nutrients" to conform to the citation style.

The authors can improve the way they present the results, their results are not distinguished from the literature, because they have directly discussed their results.

My comments are listed below:

Line 72 : specify the solution concentration

Line 77 : and in the whole document, for the chemical formulas the numbers in index.

Line 95 : use the journal citation system to uniformize your references in the whole document.

Line 158 : spacing of numbers and units ;

Line 168 : separated number and symbole

Line 177 : speace between number and unit

Line 180 : never begin a sentence with %.

Line 196 : use the journal font style throughout your manuscript

Line 235 : delete the dot

Line 246 : To remove the point, it is unitil

Line 283 : use a recent journal article to cite all the figures in the manuscript

Line 294 : review the notation of the ions

Line 329 : and in the whole document, use a recent journal article for the citation of figures. it is necessary to standardize ;

Line 332 : please space out the units; example: 3005.29±427.80μg/mL should be 3005.29±427.80 μg / mL.

Line 399 : delete the dot after the reference

Line 417 and 433 : H2O2 instead of H2O2 an O2 instead of O2

Line 426: What information do you mean by "to"?

Line 450 : please separate "value" with the number 1

Figure 5 : please review how to insert the figures of speech according to the journal by referring to a newly published article ;

Line 513 : Please write sub-G1 phase instead of sub-G1phase

Line 520 : Please write BCl-2 instead of Bcl-2

Line 541 : What is the difference between h and hrs? If it's the same meaning, it should be harmonised ;

Line 555 : separate extract and reference [37] ;

The list of references is to be revised in its entirety to conform to the style of the journal.

Author Response

In general, the theme approached by the authors in this study is very interesting and topical. the research of nutraceuticals is a subject that deserves more investigation because of its interest for the sustainable human development. However, this work in this form does not facilitate the comprehension of the thematic.

The introduction can be improved;

We believe that the introduction is concisely written and that the research question and study objectives are clearly defined in the introduction.

The authors should review the way they cite figures in the body of the text;

The figures were modified and cited according to the journal instructions

The bibliographic references should be revised according to the standards of the journal.

All references changed according to the journal instructions

Authors can use a recent article published in "Nutrients" to conform to the citation style.

Recently published articles in Nutrients were used to conform to the citation style

The authors can improve the way they present the results, their results are not distinguished from the literature, because they have directly discussed their results.

Since this is the first work to report the effects of DSE on labile iron activity, DNA damage, and protein damage, in addition to the activities of Tyrosinase, Acetylcholinesterase, and Porcine α-amylase, therefore, it is not possible to compare the results for these indexes with the literature because they are lacking. However, the results of other indexes, such as the effects of DSE on the proliferation and apoptosis of hepatic, colorectal, and breast cancer cells, were compared with results found in the literature.

My comments are listed below:

Line 72 : specify the solution concentration

Changed from (1:1) to (1: 1, vol / vol) it’s mean for example 100 ml methanol to 100 waters

Line 77 : and in the whole document, for the chemical formulas the numbers in index.

The study materials, including all chemicals used in this research, have been included in the Methods section.

Line 95 : use the journal citation system to uniformize your references in the whole document.

Mendeley program was used to format the references according to journal format

Line 158 : spacing of numbers and units ;

All numbers and units were spaced in all the manuscript

Line 168 : separated number and symbole

All numbers were separated in all the manuscript

Line 177 : speace between number and unit

whole the manuscript were spaced between number and unit

Line 180 : never begin a sentence with %.

The sentence which begins by % changed to begin by Percentage

Line 196 : use the journal font style throughout your manuscript

The font was homogenized through whole the manuscript according to journal font

Line 235 : delete the dot

The dot was deleted

Line 246 : To remove the point, it is unitil

The point was removed

Line 283 : use a recent journal article to cite all the figures in the manuscript

Done through whole the manuscript

Line 294 : review the notation of the ions

notation of the ions was reviewed

Line 329 : and in the whole document, use a recent journal article for the citation of figures. it is necessary to standardize ;

The citation of figures in whole document were standardized

Line 332 : please space out the units; example: 3005.29±427.80μg/mL should be 3005.29±427.80 μg / mL.

Spacing in whole manuscript done

Line 399 : delete the dot after the reference

The dot after the reference was removed

Line 417 and 433 : H2Oinstead of H2O2 an Oinstead of O2

Done

Line 426: What information do you mean by "to"?

Started or ranged from No. till or to No.

Line 450 : please separate "value" with the number 1

Separated

Figure 5 : please review how to insert the figures of speech according to the journal by referring to a newly published article ;

Done through the whole manuscript

Line 513 : Please write sub-G1 phase instead of sub-G1phase

Done

Line 520 : Please write BCl-2 instead of Bcl-2

Done

Line 541 : What is the difference between h and hrs? If it's the same meaning, it should be harmonised ;

h was changed to hrs through the whole manuscript

Line 555 : separate extract and reference [37] ;

Separated

The list of references is to be revised in its entirety to conform to the style of the journal.

References revised up to journal style

Reviewer 2 Report

The work presented the labile iron, enzyme, cancer cell activity, and DNA and protein damage inhibitory effect of phenol-rich date palm fruit seed extract. The results have been presented well however the article need extensive editing before considered for publication.

- There are many type, grammar and spacing errors which should be corrected.

- The reference citations are not according to the Journal style, please check.

- The author's objective of the study was to be the first to report the positive effect, rather than to report the health functional effects, please rewrite it.

- The material and methods section is very confusing, by ready this section it gives an impression that the study was only done to check the above mentioned activities of commercial vitamin C, Rutin and DSE. Please check the rewrite this section to make it clear for the readers to understand. 

- The authors separated date seeds fine powder into two fractions using 0.5 mm and 0.25 mm and then used only 0.25 mm fine powder was further used for extraction, any reasoning for this?

- Why do the authors preferred to use water: method (1:1) ratio for extraction, any reference?

- There is no information provided how the extract was prepared.

- The abbreviations such as DSE should be explained in full in the first instance they appear and then abbreviated throughout the article.

- On several occasions DSE has been mentioned a nutrient, whereas upto my understanding it was an extract??

- All different tests have used the same equation 1, which is for DPPH activity, how it is possible?

- Abs test mentioned on line 99 is not present in the equation 1.

- The materials and methods lacks the information about the replications of this study.

- Figure 5 is very confusing to understand, please use different numbering within the figure

- Authors have stated that their seed extract has anti HepG-2, Caco-2, and MDA activity however only microscopic images of only anti HepG-2 cells are provided, please provide the histograms of other cells as well.   

Author Response

The work presented the labile iron, enzyme, cancer cell activity, and DNA and protein damage inhibitory effect of phenol-rich date palm fruit seed extract. The results have been presented well however the article need extensive editing before considered for publication.

The whole manuscript has been extensively edited.  

- There are many type, grammar and spacing errors which should be corrected.

All type, grammar and spacing errors through the whole manuscript were corrected.

- The reference citations are not according to the Journal style, please check.

All references have been cited according to the journal style.

- The author's objective of the study was to be the first to report the positive effect, rather than to report the health functional effects, please rewrite it.

The statement has been revised in the manuscript according to the suggestion of the reviewer.

- The material and methods section is very confusing, by ready this section it gives an impression that the study was only done to check the above mentioned activities of commercial vitamin C, Rutin and DSE. Please check the rewrite this section to make it clear for the readers to understand. 

The following statement has been added in the materials section to clarify the use of vitamin C and rutin:

“The biological activities of DSE investigated in this study were compared with the activities of vitamin C and rutin, which are well-established potent antioxidants.”

- The authors separated date seeds fine powder into two fractions using 0.5 mm and 0.25 mm and then used only 0.25 mm fine powder was further used for extraction, any reasoning for this?

Because the separation rate and efficiency of extraction of fine powders increase when the fine powder surface area is decreased

- Why do the authors preferred to use water: method (1:1) ratio for extraction, any reference?

This enables the separation of the hydrophobic molecules and hydrophilic molecules from the Date seed powder.

 References were added.

- There is no information provided how the extract was prepared.

The extract prepared information was provided

- The abbreviations such as DSE should be explained in full in the first instance they appear and then abbreviated throughout the article.

The abbreviations of DSE have been explained in the first line of the abstract   

- On several occasions DSE has been mentioned a nutrient, whereas up to my understanding it was an extract??

DSE is indeed an extract, however, it contains many nutrients as shown by previous work, however, the mention of DSE as a nutrient per se has been eliminated in the manuscript.  

- All different tests have used the same equation 1, which is for DPPH activity, how it is possible?

From the below equations it’s clear that all these tests use the same principle of calculating the % inhibition. Before all equations were written but the journal reduced it to eq. 1 considering it unnecessary to repeat the equations. Also, there are many papers below that use the eq. in the same way 

% Inhibition DPPH = Abs. control – Abs. sample X 100                                                                                                                                                                          Abs. control

% Inhibition ABTS = Abs. control – Abs. sample X 100                                                                                                                                                                          Abs. control

% Inhibition NO = Abs. control – Abs. sample X 100                                                                                                                                                                               Abs. control

% Inhibition Porcine α-amylase = Abs. control – Abs. sample X 100                                                                                                                                                                      Abs. control

% Inhibition Tyrosinase = Abs. control – Abs. sample X 100                                                                                                                                                                   Abs. control

% Inhibition Acetylcholinesterase = Abs. control – Abs. sample X 100                                                                                                                                                                  Abs. control

Hosam M. Habib, Esmail M. El Fakharany, E. Kheadr, and Wissam H. Ibrahim Grape seed extract reduces the proliferation of liver, colon, and breast cancer cells and exerts antioxidant, enzymes, DNA, BSA damage, labile iron inhibitory activities, and cytotoxicity. scientific reports, 12:12393 (2022) https://doi.org/10.1038/s41598-022-16608-2

Hosam M. Habib, E. Kheadr, and Wissam H. Ibrahim. Inhibitory effects of honey from arid land on some enzymes and protein damage. Food Chemistry 364 (2021) 130415, https://doi.org/10.1016/j.foodchem.2021.130415

Hosam M. Habib, Serah W. Theuri, Ehab E. Kheadr Fedah E. Mohamed, (2017). DNA, BSA damage inhibitory activities, anti-acetylcholinesterase, anti-porcine α – amylase and antioxidant properties of Dolichos lablab bean. Food & Function, 2017, 8, 881. http://dx.doi.org/10.1039/c6fo01164k

Hosam M. Habib, Fatima T. Al Meqbali, Hina Kamal, Usama D. Souka, Wissam H. Ibrahim. (2014). Bioactive components, antioxidant and DNA damage inhibitory activities of honeys from arid regions. Food Chemistry 153 (2014) 28– 34. http://dx.doi.org/10.1016/j.foodchem.2013.12.044

Hosam M. Habib, Wissam H. Ibrahim, 1 Regine. Schneider-Stock, Hassan M. Hassan. (2013) Camel Milk Lactoferrin Antiproliferative in colorectal cancer cells, Antioxidant and DNA Damage Inhibitory Activity, Food Chemistry, Food Chemistry 141 (2013) 148–152 http://dx.doi.org/10.1016/j.foodchem.2013.03.039

- Abs test mentioned on line 99 is not present in the equation 1.

Changed to Abs. control test

- The materials and methods lacks the information about the replications of this study.

The replications in this study mentioned on the materials and methods under the statistical analysis ((The statistical analyses were carried out in triplicate (n =3). Also mentioned at the tables and figures

- Figure 5 is very confusing to understand, please use different numbering within the figure

Figure 5 was splatted to 2 figures (figure 5) (figure 6)

- Authors have stated that their seed extract has anti HepG-2, Caco-2, and MDA activity however only microscopic images of only anti HepG-2 cells are provided, please provide the histograms of other cells as well.   

We provide only the image of anti HepG-2 as example only for the treatment. Because of when provided all images for Caco-2 and MDA that make the figure and the discussion more complicated and confusing to understand by reader, as we did in our previous publication ((Habib, H.M.; El-Fakharany, E.M.; Kheadr, E.; Ibrahim, W.H. Grape Seed Proanthocyanidin Extract Inhibits DNA and Protein Damage and Labile Iron, Enzyme, and Cancer Cell Activities. Sci. Rep. 2022, 12, 12393, http://dx.doi.org/10.1038/s41598-022-16608-2.))

Round 2

Reviewer 1 Report

I actually think the authors really improved the article and the article is of enormous scientific interest. Language and style are fine, only minor spell check required.

Reviewer 2 Report

The authors have addressed most of my concerns.